# Geographic information system and information visualization capacity building: Successful polio eradication and current and future challenges in the COVID-19 era for the World Health Organization's African region

**John Kapoi Kipterer**[1]*, **Kebba Touray**[1], **Akpan Ubong Godwin**[1], **Aboubakar Cisse**[1], **Babona Nshuti Marie Aimee**[1], **Ngobe Busisiwe**[1], **Chefor Ymele Demeveng Derrick**[1], **Green Hugh Henry**[1], **Ndoutabe Modjirom**[1], **Vince Seaman**[2], **Jamal Ahmed**[1]

**1** World Health Organization—Regional Office for Africa, Cité du Djoué, Brazzaville, Republic of the Congo,
**2** Bill & Melinda Gates Foundation, Seattle, Washington, United States of America

* jkipterer@who.int, karantili@gmail.com

## Abstract

Despite a half-century-long global eradication effort, polio continues to have a devastating impact on individuals and communities worldwide, especially in low-income countries affected by conflict or geographic barriers to immunization programs. In response, the World Health Organization (WHO) Global Polio Eradication Initiative (GPEI) employs disease surveillance and vaccination campaigns coordinated through the WHO Regional Office for Africa (AFRO) Geographic Information System (GIS) Centre. Established in 2017, the AFRO GIS Centre played a key role in the eradication of wild-type polioviruses (WPVs) in 2020, but the COVID-19 pandemic, emergence of circulating vaccine-derived polioviruses, and transmission of WPV1 from Central Asia have led to a resurgence of polio in Sub-Saharan Africa. The AFRO GIS comprises a set of mobile device or cloud-based tools for geospatial data collection, analysis, and visualization. Using tools such as Auto-Visual Acute Flaccid Paralysis Detection and Reporting, electronic surveillance, and Integrated Supportive Supervision, GIS personnel collect polio case numbers and locations, track field worker activities, follow the movements of nomadic populations vulnerable to polio and other diseases, and determine needs for further healthcare deployments. The system is location specific and operates in real time, enabling the AFRO GIS to promptly target its responses to polio, COVID-19, Ebola virus disease, and other public health crises and natural disasters. The present review describes the components of the AFRO GIS and how the AFRO GIS Centre coordinated on-the-ground polio eradication efforts to help secure Africa's certification as WPV free. It also examines current and prospective challenges regarding other disease outbreaks in the COVID-19 era and how the AFRO GIS Centre is addressing these ongoing public health needs.

**Data Availability Statement:** The data that support the findings of this study are openly available from the WHO/AFRO GIS Centre at: https://afro-gis-center-who.hub.arcgis.com/pages/training, https://geopode.world/ and https://who.maps.arcgis.com/apps/dashboards/4d1662750a4849b0af606cd83b662938#mode=edit. The authors confirm that the data supporting the findings of this study are available within the article.

**Funding:** Funding for the WHO AFRO GIS is provided by the World Health Organization, Geneva, Switzerland, and Bill & Melinda Gates Foundation, Seattle, Washington, USA. The Bill & Melinda Gates Foundation provided human resource support during the trainings. Both funders were involved in study design, data collection and analysis, decision to publish, and preparation of the manuscript.

**Competing interests:** The authors have no competing interests to declare.

## Introduction

Paralytic poliomyelitis (polio)—an incurable and potentially fatal disease—persists in pockets throughout the world, despite a global eradication effort underway for more than half a century [1, 2]. Wild-type polioviruses (WPVs) 2 and 3 were declared globally eradicated in 2015 and 2019, respectively [1], and the World Health Organization (WHO) African region was declared WPV2 free in 2020 [3]. In addition, the emergence of circulating vaccine-derived polioviruses (cVDPVs) has led to increasing case reports of acute flaccid paralysis (AFP)—the primary sign of polio infection—throughout the globe. Over 3000 polio cases have been reported worldwide since 2020, including nearly 700 cases in the 12 months prior to March 2023 (**Fig 1**) [4–6]. Although polio case counts are highest in low-income countries affected by war or geographic barriers to immunization programs, polio has also been detected in a growing number of middle- and high-income nations [6].

Over 95% of polio cases reported in 2022 were in African countries, including a small, but significant number caused by a Pakistani WPV1 lineage documented in Mozambique and Malawi [7], and a large majority caused by cVDPVs [6]. Vaccine-derived polio arises in under-immunized populations due to genetic instability of the live-attenuated viruses included in the oral polio vaccine (OPV). One of the strengths of OPV is that the attenuated virus can spread to unvaccinated contacts of vaccine recipients, immunizing them against polio and extending protection to the larger community. In areas with low vaccination rates, however, circulating OPV viruses sometimes mutate, reverting to neurovirulent cVDPV strains that produce paralysis similar to that seen in WPV infections [8, 9].

Interrupting transmission of both WPV1 and cVDPVs are the primary goals of the Polio Eradication Strategy 2022–2026 developed by the WHO Global Polio Eradication Initiative

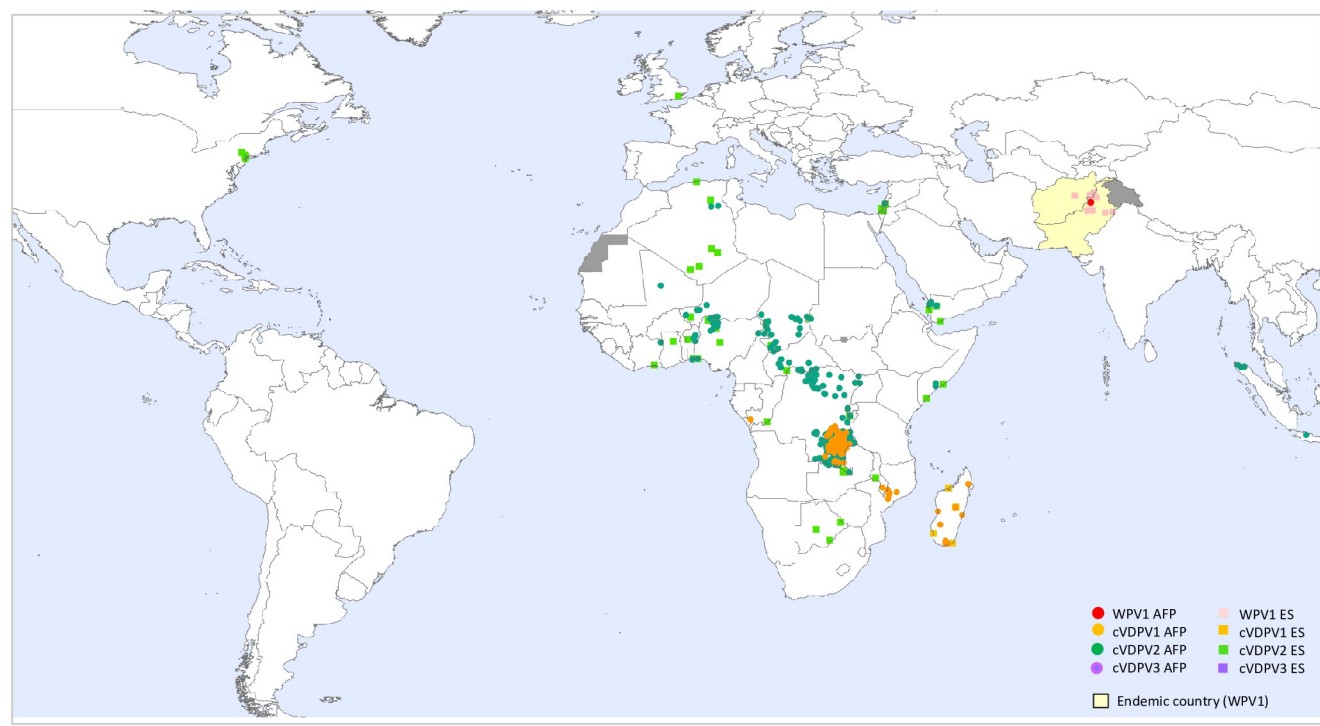

**Fig 1. Global case counts of wild-type poliovirus 1 (WPV1) and circulating vaccine-derived polioviruses (cVDPVs) 1, 2, and 3 reported between 29 October 2022 and 30 October 2023.** Counts exclude viruses detected from environmental surveillance. Source: World Health Organization/HQ/Polio Eradication Program @2024 [6].

(GPEI) [10]. Along with developing partnerships with political entities and affected communities, the WHO GPEI uses a 2-pronged approach involving intensive disease surveillance and campaigns to boost vaccination coverage above 95% (the threshold at which cVDPV transmission is eliminated) [10, 11]. These tactics have successfully stopped past outbreaks, but the COVID-19 pandemic interfered with GPEI efforts, promoting a resurgence in polio cases [10]. The resurgence of WPV1 into the African continent and ongoing outbreaks of cVDPV infections highlights the need for highly coordinated and intensive polio surveillance and immunization efforts in Africa, including deployment of novel OPV type 2 (nOPV2), which reduces the risk of cVDPV2 (the most common form of vaccine-derived polio) [10, 12]. The rollout of nOPV2 in the African region is aimed at addressing the outbreak of polio, strengthening immunization efforts, reducing the transmission of cVDPV2, and contributing to the efforts of the GPEI [13, 14]. This initiative represents a significant milestone in combating polio outbreaks, particularly in the countries of the Lake Chad Basin region, which have been significantly impacted by the disease [15]. The introduction of nOPV2 in Nigeria and Liberia is a crucial step towards effectively responding to the ongoing cVDPV2 outbreaks and preventing further transmission of the virus [16].

In Sub-Saharan Africa, with its limited access to geographic information system (GIS) infrastructure, coupled with inadequate technical and analytical skills [17], GPEI activities are coordinated through the WHO Regional Office for Africa (AFRO) GIS Centre. Since its establishment in 2017, the AFRO GIS Centre has played a vital role in achieving Africa's WPV-free status. After the COVID-19 pandemic, the GIS Centre also supported the development of electronic tools for data collection, analysis, and monitoring that have been used not only in polio, but in COVID-19, Ebola virus disease, and other public health initiatives throughout the region [18].

This review describes the components of the WHO AFRO GIS and how the AFRO GIS Centre coordinated on-the-ground polio eradication efforts to help secure Africa's certification as WPV free. It also examines current and prospective challenges in the COVID-19 era and how the GIS Centre is addressing these ongoing public health needs in Africa.

## Geographic information system overview

In the generic sense, GIS is a computer system used to capture, store, verify, and display 3-dimensional location data, ie, data related to positions on Earth's surface [19]. In the public health sphere, GIS is used to integrate and track data collected for environmental and disease outbreak surveillance, track field activities such as vaccination campaigns, and provide essential information on the movement and activities of field personnel [18, 20].

## WHO AFRO geographic information system

The WHO AFRO GIS consists of multiple, mobile- and cloud-based software tools for geospatial data collection, analysis, and visualization, which support polio and other public health interventions at regional, national, and local levels in the African region [20]. These tools enable the WHO AFRO GIS Centre to address public health needs such as determining which regions need more healthcare workers, the location of vulnerable populations and their level of healthcare access, and the optimal deployment of mobile clinics [18]. Because GIS is location specific and operates in real time, it facilitates prompt and focused responses to health crises and natural disasters.

## Solutions of the WHO AFRO geographic information system

Mapping and analytic software programs comprising the WHO AFRO GIS Centre's solutions include Power BI (Microsoft, Redmond, Washington, USA), a business intelligence

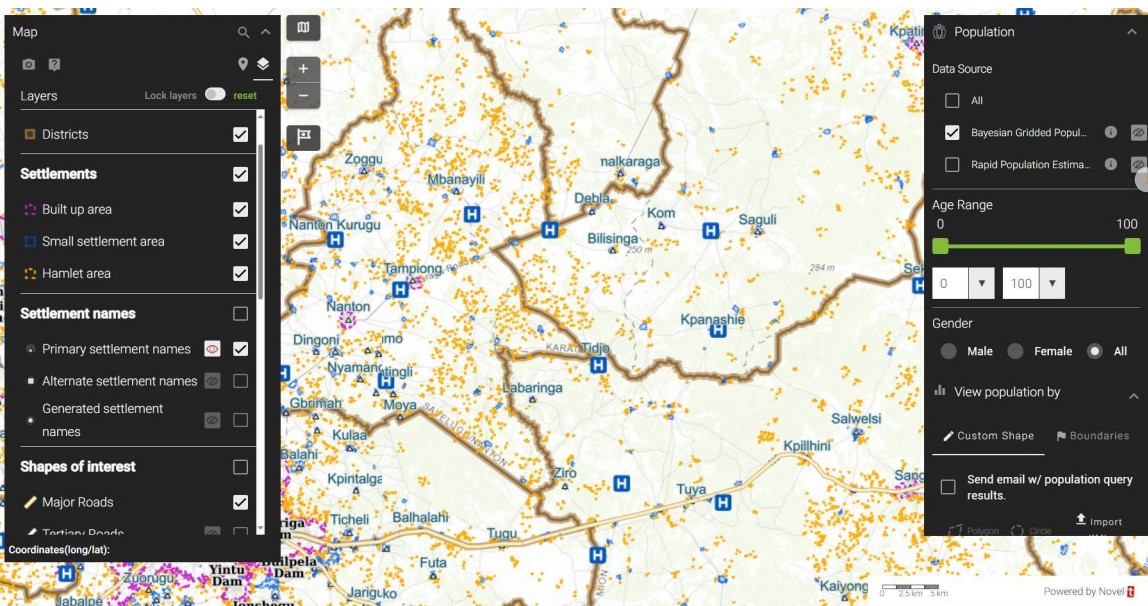

**Fig 2. Screen capture from Ghana (northern region) GeoPoDe display showing location of health facilities and towns.** The data are obtained by WHO from the ministry of health and bureaus of statistics, and are incorporated in the GeoPoDe by the WHO in support of ministry of health decision-making. Source: Bill and Melinda Gates Foundation 2022 [23].

visualization tool; ArcGIS Online (Esri, Redlands, California, USA), an open data kit for mobile application tools, which is a cloud-based data mapping and collection program [21]; Vaccination Tracking System (VTS) [22]; and Geospatial Tracking System (GTS). The VTS and GTS are deployed to support the tracking of vaccination teams during polio outbreak campaigns. In addition, the AFRO GIS Centre, in collaboration with Novel-t (Geneva, Switzerland), uses GeoPoDe (Geographic, Population, & Demographic Data; Bill & Melinda Gates Foundation, Seattle, Washington), an open-source repository for geospatial reference data sets. GeoPoDe shows population data on topographic maps that also highlight the user's choice of points of interest, such as settlements, schools, and health facilities (**Fig 2**), and can be used to support microplanning activities [23].

These tools permit GIS personnel to collect AFP case numbers and locations, and data for environmental surveillance, track field activities, and identify nomadic population settlements s. In addition, data collected by field personnel using mobile applications such as Auto-Visual AFP Detection and Reporting (AVADAR), electronic surveillance (eSURV), and Integrated Supportive Supervision (ISS) are used to not only identify suspected AFP cases and polio immunizations, but to determine needs for further healthcare service deployment (**Figs 3 and 4**) [24].

Integrated Supportive Supervision and eSURV are mobile health solutions for effective supervision and surveillance in healthcare. Integrated Supportive Supervision utilizes electronic checklists on Android phones for active case finding and routine immunization supervision by the WHO and government staff [26]. Electronic surveillance ensures government surveillance agents conduct active searches using mobile phones in health facilities and communities [27]. Auto-Visual AFP Detection and Reporting is a community-based surveillance solution that employs the use of a mobile app combining visual, audio, and Short Message Service reporting to enhance AFP surveillance [25]. It aims to improve AFP case detection and reporting, enhance reporting timeliness, ensure "Zero" reporting of AFP cases, provide real-

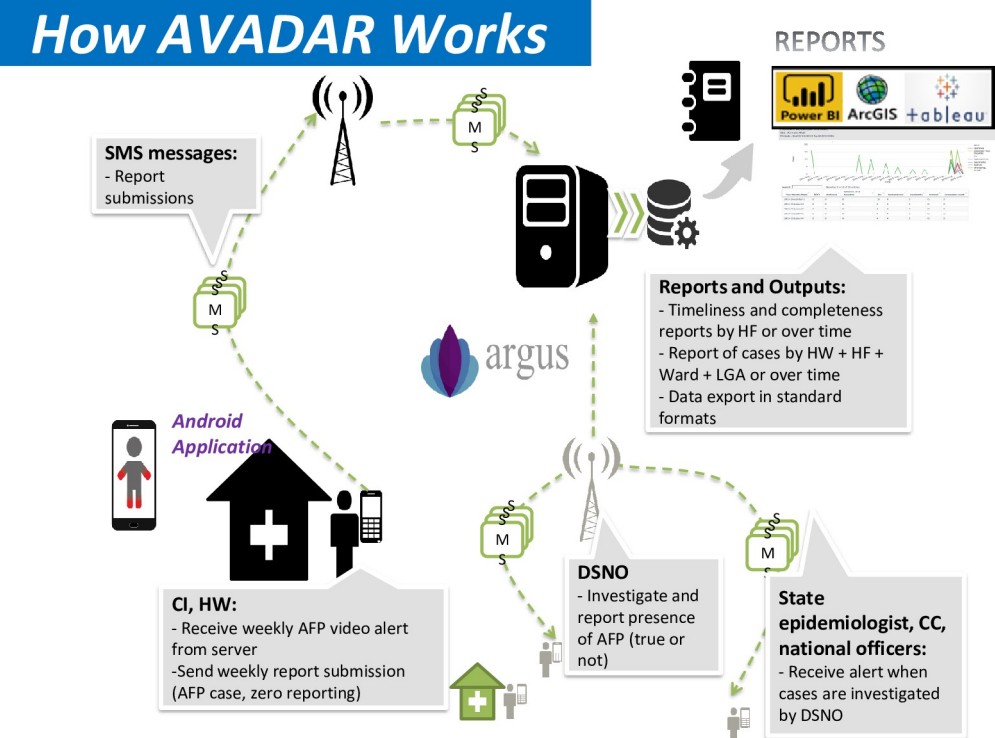

**Fig 3. Auto-Visual AFP Detection and Reporting (AVADAR).** AFP, acute flaccid paralysis; CC, cluster consultant; CI, community informant; DSNO, Disease Surveillance Notification Officer; HF, health facility; HW, health worker; LGA, Local Government Area; ODK, Open Data Kit; SMS, short message service [25].

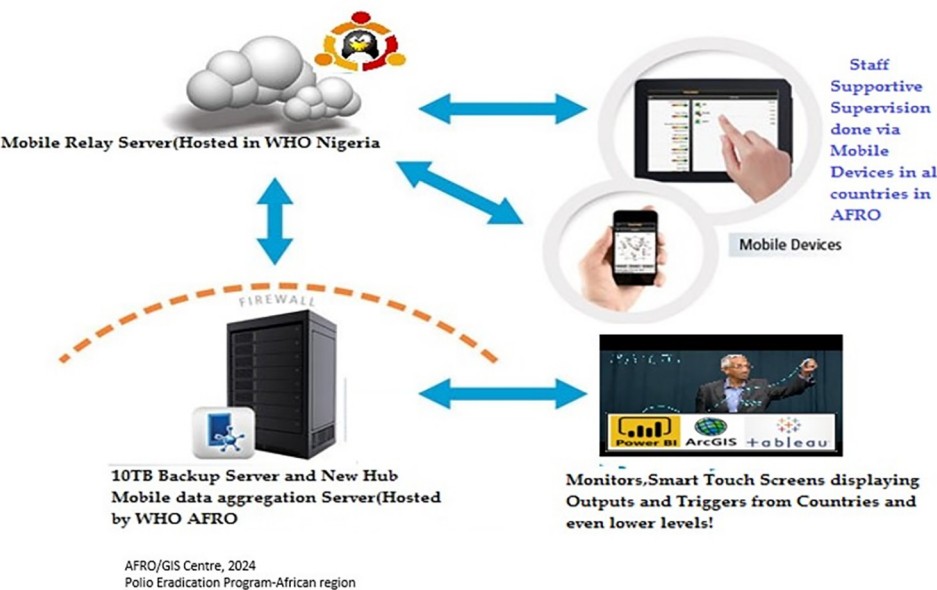

**Fig 4. Electronic surveillance (eSURV).** AFRO, World Health Organization (WHO) Regional Office for Africa. Source: AFRO/GIS Centre 2024; Polio Eradication Program-African region.

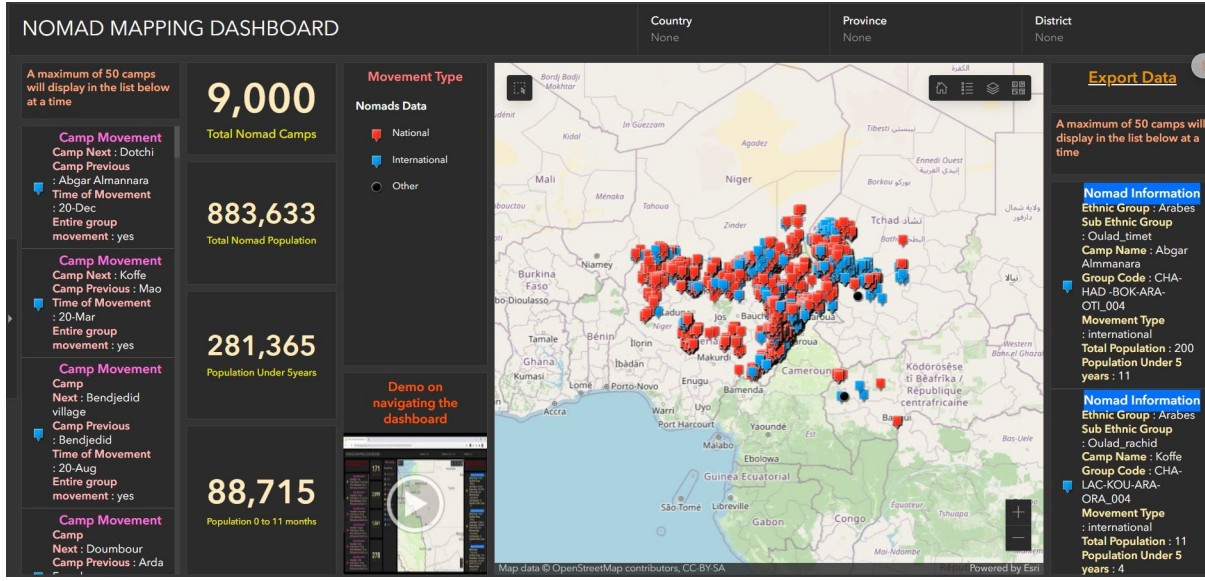

**Fig 5. Chad Lake Basin nomad mapping dashboard.** Location data for nomadic populations are used in immunization planning to ensure healthcare workers are able to reach nomadic communities. Source: World Health Organization Regional Office for Africa Geographic Information System Centre [28].

time case information, and increase community participation and ownership in surveillance efforts.

The data collected and stored in the WHO servers through these solutions are connected to ArcGIS Online and Power BI for analyses that involve mapping and visualizations, with program indicators of surveillance visualized in a dashboard. Visualization analytics are used by the program staff to identify gaps in surveillance, where areas not visited by active case search and routine surveillance personnel are quickly identified for action, while the AFP cases reported by AVADAR are also integrated in the AFP database for follow-up.

The WHO AFRO GIS Centre staff can also analyze country-level outbreak responses, surveillance, and immunization efforts in real time on a web-based dashboard (**Fig 5**) [18, 28]. Based on data and information collated from these multiple platforms, the AFRO GIS Centre develops information visualization platforms such as the Africa Rapid Response Team Hub to provide timely access to information for informed decision-making.

## WHO AFRO geographic information system capacity-building efforts: 2017–2022

**2017–2019.** Proof of concept for the application of GIS software to polio eradication programs was documented in multiple African nations prior to the WHO AFRO GIS Centre's establishment in 2017 [22, 29, 30]. To build on these successes, the AFRO GIS Centre was established to ensure proper national adoption of GIS, build the required capacity in relevant knowledge areas at all levels throughout Africa, and support regional and national interventions in surveillance, routine immunization, supportive immunization activities, information management, monitoring, and evaluation and accountability of teams and individuals at all levels [20]. To achieve these goals, the AFRO GIS Centre rolled out a series of train-the-trainer initiatives consisting of regional, in-person workshops intended to teach GIS focal points (ie, national health ministry officials and data managers, as well as the WHO data management representative in each country) how to use GIS for field data collection, management, and

visualization, and deploy GIS resources locally. Initially, AFRO GIS Centre efforts were directed toward polio eradication in Africa. After WPV-free certification was achieved, AFRO GIS Centre initiatives were focused on improving African member states' ability to use AFRO GIS Centre resources to address cVDPV outbreaks, Ebola, COVID-19, and other emerging crises.

After the WHO AFRO GIS Centre was established, use of GIS mobile applications in community-based surveillance activities such as AVADAR, and surveillance active cases such as eSURV and ISS helped close gaps in polio surveillance and immunization in several high-risk areas, including communities in Burkina Faso, the Lake Chad region, Mali, and Niger [20]. One of the most dramatic AFRO GIS Centre successes involved improving polio surveillance and immunization coverage in the remote Lake Chad region [20, 31]. The WHO AFRO GIS teams from Niger successfully vaccinated >200,000 children from Lake Chad island communities that had frequently been left out of previous surveillance and immunization efforts [31–34]. In addition in 2019, an estimated 3%–5% of African WPV cases occurred in ~35,000 nomads living in the Lake Chad Basin, which includes territory belonging to Cameroon, Central African Republic, Chad, Niger, and Nigeria. Using GIS mobile data collection tools, field personnel were able to identify specific nomadic groups, document their movements, and conduct AFP case surveillance to support planning and decision-making [20]. In the past 3 years despite numerous challenges, including armed conflicts in the region, >40,000 children in 3451 nomad camps representing 13 ethnic groups in 62 high-risk districts have been immunized against polio and other communicable diseases. These groups continue to be monitored and evaluated (**Fig 5**) [18].

Despite these successes, a 2019 survey assessing GIS adoption by the 47 WHO AFRO member states revealed deficiencies in GIS capacity throughout the continent. As shown in **Fig 6**, most countries lacked updated health facility master lists (comprehensive listings of the names, locations, and types of health facilities in each country); adequate office space or resources for GIS personnel, equipment, or software; or sufficient GIS staffing or training [20]. Where GIS training was provided, it failed to reach frontline workers. In addition, approximately 90% of GIS training required travel, 24% of which was international.

During an October 2019 meeting of the AFRO GIS Summit in Brazzaville, Republic of the Congo, representatives of the 47 member states reviewed WHO AFRO GIS Centre activities and identified barriers to polio eradication progress, including poor coordination between the WHO and national health ministries, inadequate preparedness of local health workforces for GIS, lack of interoperability, and inadequate sustainable financing [20]. To address the identified gaps and barriers, summit participants developed a list of recommendations for individual WHO AFRO member states and the AFRO GIS Centre (**Table 1**).

**2019 to present.** Since the AFRO GIS Summit, GIS training and software capabilities have been updated and extended. The WHO/AFRO GIS Centre staff train data managers from ministries of health and WHO country offices, who, in turn, cascade the training to the subnational level, ie, district-level staff. Train-the-trainer workshops now are conducted to fulfil the following objectives:

- Train participants as the trainers of trainers so that they can develop relevant content for country-level training and support;

- Build the capacity of participants in the use of the Power BI visualization tool and ArcGIS Online mapping software for standard and quality map production for country programs and outbreak response;

- Train participants on web-mapping applications provided in ArcGIS Online tools for dynamic mapping to improve information dissemination for decision-making;

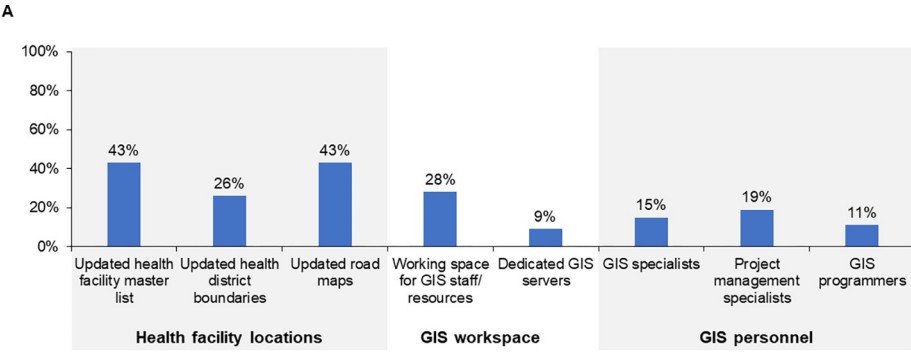

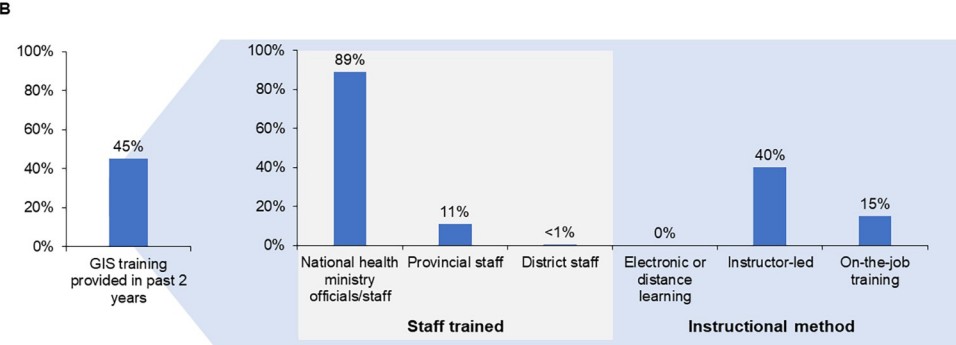

**Fig 6. Selected findings from 2019 national assessment of geographic information system (GIS) adoption in Africa (n = 47 World Health Organization African Region member states)** [20]. *A*: Survey responses from WHO AFRO GIS country teams regarding documentation of healthcare facility locations; provision of office resources for GIS staff, equipment, and software; and availability of trained GIS staff. Survey questions focused on geographic details, personal information, foundations of GIS, baseline data and standards, capacity building, services and applications, evaluation, challenges of GIS implementation, and lessons learned. *B*: Types of staff trained and instructional methods used in countries where GIS training was provided in last 2 years.

**Table 1. Summary of key recommendations for WHO member states and WHO from the 2019 African Regional GIS Summit** [20].

| WHO MEMBER STATES | WHO |
|---|---|
| • Foster interministry collaboration to ensure sharing of information and resources, and develop comprehensive and all-inclusive digital-health strategies<br>• Integrate GIS into country policies, strategies, and action plans<br>• Ensure health-system stakeholders and partners share outputs of their activities, and are aware of and have access to GIS<br>• Support interoperability, information sharing, and decentralized data management to ensure availability and accuracy of GIS to be used in emergencies<br>• Develop and regularly update master lists of health facilities and population settlements<br>• Maintain maps with up-to-date subnational boundaries<br>• Determine definitions and levels of health facilities, and treatment and service-delivery sites | • Provide technologic and technical support to member states on data management and information use, and national strategies, plans, funding, and GIS implementation<br>• Develop and foster mentorship programs to facilitate dissemination of GIS-related knowledge and resources<br>• Provide guidance and documentation on GIS tools and systems<br>• Develop workspace for data collection, validation, and sharing<br>• Develop repository of minimal sets of standards and standard operating procedures<br>• Develop platform for member states and partners to document and share experiences, success stories, and details of GIS-related national projects and innovative technologies |

GIS, geographic information system

- Orient participants on the use of WHO AFRO GIS Centre innovative solutions and platforms, including GeoPoDe; and

- Train participants on the use of the open data kit and ArcGIS Survey 123 to create maps and other survey tools, and deploy them in AFRO GIS Centre-managed servers

In addition, GIS training provides instruction on tools developed to support nOPV2 deployment and polio surveillance data analysis, including outbreak/cluster analysis and campaign coverage estimation. Trainees learn about the collection, validation, and use of geospatial reference datasets and polio spatial data used in AFP case and environmental surveillance, as well as immunization campaigns and nomadic population tracking. They are also provided with an introduction to spatial data analysis, the principles of spatial epidemiology, and the use of GTS for outbreak response campaigns and other activities.

After the latest series of GIS capacity-building workshops held over the first 6 months of 2022, the GIS network now consists of >120 GIS focal points across 45 of 47 countries in the WHO African region and >200 health ministry workers have been trained on GIS-related mobile health skills. In addition, 20 African nations (18 WHO AFRO member states) have made their population and demographic data available on GeoPoDe based on 2020 WorldPop datasets [23, 35]. The WHO AFRO GIS enables frontline workers to promptly alert health ministries of potential disease outbreaks based on real-time data from active AFP surveillance visits and environmental sampling. The latter, which involves the systematic sampling of sewage and testing for the presence of WPVs and cVDPVs, has been growing in importance as an early warning system for impending outbreaks, as it may allow the detection of poliovirus in a community before any AFP cases are reported [36]. When alerted to a potential outbreak, health ministry officials can use GIS to evaluate surveillance data alongside population estimates and locations, and track response team movement and activities, including in real time [5]. When alerted of an outbreak, WHO rapid response teams mobilize to affected areas within 48–72 hours [24]. Thus, the AFRO GIS provides WHO focal points with the means of more rapidly and efficiently implementing the WHO outbreak response, including the deployment of nOPV2 where appropriate.

## Use of the WHO AFRO geographic information system beyond polio

The integration of polio transition processes involves the bringing together of various stakeholders and resources to ensure a smooth and effective transition. This includes integrating polio surveillance systems into existing disease surveillance networks, incorporating polio vaccinations into routine immunization programs, and merging the expertise and infrastructure developed for polio eradication with broader public health initiatives [37]. During the COVID-19 pandemic, WHO AFRO GIS Centre systems such as eSURV and ISS were repurposed for COVID-19 surveillance and contact tracing. The modified systems included contact-tracing capabilities, in addition to surveillance data collection specific to the identification of SARS CoV-2 infections; collected data could be viewed and analyzed using visualization platforms to provide health officials with actionable insights. After a pilot program conducted between April and June 2020, the COVID-19 contact-tracing application was initially deployed in Benin, Cameroon, Nigeria, South Sudan, Uganda, and Zimbabwe. A surveillance application designed for use in health facilities was adopted by 27 Sub-Saharan African countries. Because health officials were already familiar with GIS for polio surveillance, minimal training was required for COVID-19 application. In addition, the platform was designed to allow connectivity with existing contact-tracing and facility-based surveillance systems [38]. In Niger, integration of the ISS system with the COVID-19 contact-tracing application enabled remote education of frontline workers on COVID-19 recognition and safety protocols, in addition to

the detection of COVID-19 cases and contact tracing in the community [39]. Environmental sampling for SARS-CoV-2 is also an important early detection tool for potential COVID-19 outbreaks that could be facilitated using GIS resources [40].

Another important effort currently underway is the development of a shared, centralized WHO AFRO geodatabase that will include information on health facilities, administrative boundaries, environmental and disease surveillance sites and data, and other GIS information in a centralized portal. For example, the WHO AFRO GIS is working with member states' ministries of health to create a consolidated health facility master list for AFRO. When WHO field staff conduct immunizations or investigate infectious disease cases reported at health facilities or traditional healer sites, they collect location data for those sites (eg, name, location, and type of facility) through use of the eSURV and ISS mobile application. The inclusion of traditional healers in the master list is an important component of polio disease surveillance because they provide healthcare to most Africans, especially in rural areas [41]. The WHO AFRO GIS Centre will incorporate the collected information into health facility lists provided by each country's ministry of health to update and validate the data in the GIS health facility master list. When complete, the health facility master list will exist as a living document and resource for WHO staff and their partners in each AFRO member state.

### Future outlook on WHO AFRO geographic information system capacity

To address ongoing needs for GIS software development, support, and training, the WHO has partnered with Esri (the developer of ArcGIS) and other groups. These partnerships will facilitate not only training on GIS software, but also continue to lead to improved automated mapping support, data visualization, and analysis tools. In addition, the WHO-Esri collaboration will facilitate the development of advanced training modules covering advanced analytics for statistics and spatial analysis.

Ongoing efforts to improve WHO AFRO GIS capabilities include the need to ensure adequate GIS server capacity. The lack of dedicated servers at the country level was one of the largest gaps identified during the 2019 AFRO GIS Summit [20]. As GIS software continues to evolve, incorporating more layered and complex datasets, the need for a robust hardware infrastructure to support data visualization and processing will continue to grow. Addressing the need for server capacity is one of the most important goals for the future. To streamline GIS and capacity building efforts across the WHO African region, the WHO AFRO GIS Centre initiated intercluster GIS collaboration for all clusters, and a GIS working group is being established with a secretariat to sustain and strengthen collaboration around GIS infrastructure capacity building and baseline data development.

### Conclusion

Geographic information systems are powerful decision-support tools for public health and were integral to the eradication of WPVs in Africa. The reemergence of WPV1 and continued outbreaks of cVDPVs, as well as gaps in immunization coverage, however, call for strengthening surveillance and implementing high-quality vaccination campaigns to achieve polio eradication. The WHO AFRO GIS Centre will continue to improve GPEI capacity in terms of trained personnel, and systems support and integration, with the goal of faster detection and response to polio and other disease outbreaks in the COVID-19 era.

### Author Contributions

**Conceptualization:** John Kapoi Kipterer, Kebba Touray, Akpan Ubong Godwin, Vince Seaman.

**Data curation:** John Kapoi Kipterer, Kebba Touray, Aboubakar Cisse, Babona Nshuti Marie Aimee, Ngobe Busisiwe, Chefor Ymele Demeveng Derrick.

**Formal analysis:** John Kapoi Kipterer, Kebba Touray, Akpan Ubong Godwin, Aboubakar Cisse, Babona Nshuti Marie Aimee, Ngobe Busisiwe, Chefor Ymele Demeveng Derrick.

**Funding acquisition:** Jamal Ahmed.

**Investigation:** Aboubakar Cisse, Babona Nshuti Marie Aimee, Chefor Ymele Demeveng Derrick.

**Methodology:** John Kapoi Kipterer.

**Project administration:** Kebba Touray, Akpan Ubong Godwin, Babona Nshuti Marie Aimee, Ngobe Busisiwe, Chefor Ymele Demeveng Derrick, Green Hugh Henry, Ndoutabe Modjirom, Vince Seaman, Jamal Ahmed.

**Resources:** Kebba Touray, Akpan Ubong Godwin, Green Hugh Henry, Ndoutabe Modjirom, Vince Seaman, Jamal Ahmed.

**Supervision:** John Kapoi Kipterer, Kebba Touray, Green Hugh Henry, Ndoutabe Modjirom, Vince Seaman, Jamal Ahmed.

**Visualization:** John Kapoi Kipterer, Kebba Touray, Akpan Ubong Godwin, Aboubakar Cisse, Babona Nshuti Marie Aimee, Ngobe Busisiwe, Chefor Ymele Demeveng Derrick.

**Writing – original draft:** John Kapoi Kipterer.

**Writing – review & editing:** John Kapoi Kipterer, Kebba Touray, Akpan Ubong Godwin, Aboubakar Cisse, Babona Nshuti Marie Aimee, Ngobe Busisiwe, Chefor Ymele Demeveng Derrick, Green Hugh Henry, Ndoutabe Modjirom, Vince Seaman, Jamal Ahmed.

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
