## [Decision Letter · Decision Letter 0]

2 Nov 2023

PONE-D-23-21989Geographic information system and information visualization: past successes and future challenges in capacity building for the World Health Organization’s African regionPLOS ONE

Dear Dr. Kipterer,

Thank you for submitting your manuscript to PLOS ONE. After careful consideration, we feel that it has merit but does not fully meet PLOS ONE’s publication criteria as it currently stands. Therefore, we invite you to submit a revised version of the manuscript that addresses the points raised during the review process.

We look forward to receiving your revised manuscript.

Kind regards,

Terna Ignatius Nomhwange, MD,DTM&H,MBA

Academic Editor

PLOS ONE

Journal Requirements:

"Funding for the WHO AFRO GIS is provided by the World Health Organization, Geneva, Switzerland, and Bill & Melinda Gates Foundation, Seattle, Washington, USA."

"Editorial support was provided by Amanda Justice and Geoff Marx of BioScience Communications, New York, New York, USA, funded by the Bill & Melinda Gates Foundation, Seattle, Washington, USA."

Funding information should not appear in the Acknowledgments section or other areas of your manuscript. We will only publish funding information present in the Funding Statement section of the online submission form. 

"Funding for the WHO AFRO GIS is provided by the World Health Organization, Geneva, Switzerland, and Bill & Melinda Gates Foundation, Seattle, Washington, USA."

5. We note that Figures 1 to 3 in your submission contain map/satellite images which may be copyrighted. All PLOS content is published under the Creative Commons Attribution License (CC BY 4.0), which means that the manuscript, images, and Supporting Information files will be freely available online, and any third party is permitted to access, download, copy, distribute, and use these materials in any way, even commercially, with proper attribution. For these reasons, we cannot publish previously copyrighted maps or satellite images created using proprietary data, such as Google software (Google Maps, Street View, and Earth). For more information, see our copyright guidelines: http://journals.plos.org/plosone/s/licenses-and-copyright.

(1) You may seek permission from the original copyright holder of Figures 1 to 3 to publish the content specifically under the CC BY 4.0 license.  

**Additional Editor Comments:**

Please review and consider all the comments and suggestions by both reviewers to provide more clarity and context within the manuscript.Both reviewers have also highlighted the focus on Polio in the current draft and the need to revise the title and text to reflect the aim of this review as stated in Lines 92-95? Is the manuscript about GIS and Polio alone or other diseases as well? You may revise the title of the manuscript accordingly showing "past" Polio Successes." and challenges "post covid" for GIS utilization for other disease outbreaks. Any examples of how the GIS system has supported any other diseases? (how is the AFRO GIS Centre addressing these ongoing public health needs?)The current title also includes "capacity building". The conclusion is specific for Polio and GPEI, is this capacity building for WHO GIS Polio teams or for country field staff for Polio and other diseases as well?In line with the recommendation on the title, please update the abstract and the conclusion to ensure alignments. Note also that the conclusion only mentions Polio and nothing about other diseases outbreaks.Lines 169-171 documents polio vaccination by WHO AFRO GIS teams? Does the AFRO GIS deploy vaccination teams or is this by the teams recruited by the Governments? Is there a reference for this?The section on use of GIS beyond Polio (Lines 245-276) is a clear example of Polio transition in action. Please include a line or two on Polio transition.Please also revise to correct sentences and punctuations as may be required for consistency E.g.:  Line 215 & 21: Train (Orient) participants "on" while Line 219: Train participants "in".

Reviewers' comments:

Reviewer's Responses to Questions

**Comments to the Author**

1. Is the manuscript technically sound, and do the data support the conclusions?

Reviewer #1: Partly

Reviewer #2: Partly

2. Has the statistical analysis been performed appropriately and rigorously? 

Reviewer #1: N/A

Reviewer #2: N/A

3. Have the authors made all data underlying the findings in their manuscript fully available?

Reviewer #1: Yes

Reviewer #2: Yes

4. Is the manuscript presented in an intelligible fashion and written in standard English?

Reviewer #1: Yes

Reviewer #2: Yes

5. Review Comments to the Author

Reviewer #1: In summary this manuscript outlines the development and expansion of the use of GIS systems in the African WHO region for primarily polio eradication. This is a very important technological development both for polio eradication and future disease surveillance and intervention monitoring. The manuscript provides narrative of the progress to date, highlighting particular GIS gaps (i.e. it is not a scientific research article). However, to a reader not involved in this GIS network, more details are required to understand the processes. The authors also present data from a 2019 survey on national GID capacity but more details are required as to who was surveyed and whether follow-up data will be collected to measure progress. It would be much clearer if different visualisations were presented for a particular outbreak that the network has worked on e.g., The authors conclude the GIS system was important for achieving certification of WPV1 in the region, it would be helpful to see some examples of the different GIS layers that contributed to this.

Minor comments:

Abstract:

‘Currently, 95% of reported polio cases are in under-vaccinated populations’ it is not clear what you mean by this given many children will not have received type 2 OPV since 2016.

Introduction

Line 49, I think you mean WPV1 free?

Line 66, By vaccine-associated do you mean vaccine-derived?

Fig 1 why do you choose this specific time period?

The introduction would benefit with a few sentences describing withdrawal of type 2 from OPV in 2016 and the following declining immunity across the region, facilitating cVDPV2 spread. A little more detail required on the rollout of nOPV2 would help the reader.

Figure 2 – Requires more details in the legend – how is the data obtained? Which part of Nigeria is shown?

Paragraph 114 – ‘Components of the WHO AFRO geographic information system’. This section is hard to follow. Perhaps include a diagram to show if any of these platforms interact? More description required to understand what ISS, eSurv and AVADAR are – how do they interact with other GIS platforms? Perhaps show an example from each? It would be helpful to highlight in the diagram what new spatial data is the GIS Centre helping to collect compared to visualising standard surveillance data. It is not clear what PowerBI is used for and what ARC GIS is used for.

Figure 3– what data is going into this dashboard? How are nomadic populations ‘tracked’ it could be interpreted as unethical tracking of certain people so please explain.

Figure 4 – what is the y axis? % missing or having. Please provide more details on the survey – questions in the appendix? Who participated?

Train the trainer – who are these trainers, how are they located across countries? Who are those being trained?

Line 232 – population and demographic data – based on what data source? Most recent census or other?

Line 262 Consolidated health facility master list – this is a very exciting development. Is it possible to report on how complete this database is? What are the barriers to obtaining more data etc? What proportion of countries have supplied data….

IPV – has the GIS Centre been involved in evaluating the (few) IPV campaigns?

Reviewer #2: Authors should align the title, the main text and the conclusion and recommendations of this paper. The paper in its current form seems to be too much focused on polio even though its intention is to highlight the application ion WHO AFRO GIS tool in prevention and control of not only polio but other diseases.

6. PLOS authors have the option to publish the peer review history of their article (what does this mean?). If published, this will include your full peer review and any attached files.

Reviewer #1: No

Reviewer #2: No

---

## [Author Response · Author response to Decision Letter 0]

28 Feb 2024

Response to Reviewer/Editor Comments 

PONE-D-23-21989

Geographic information system and information visualization: past successes and future challenges in capacity building for the World Health Organization’s African region

Response: Done.

"Funding for the WHO AFRO GIS is provided by the World Health Organization, Geneva, Switzerland, and Bill & Melinda Gates Foundation, Seattle, Washington, USA."

Response: Please add the following text re the role of the funder: “The Bill & Melinda Gates Foundation provided human resource support during the trainings. Both funders were involved in study design, data collection and analysis, decision to publish, and preparation of the manuscript.” 

"Editorial support was provided by Amanda Justice and Geoff Marx of BioScience Communications, New York, New York, USA, funded by the Bill & Melinda Gates Foundation, Seattle, Washington, USA."

Funding information should not appear in the Acknowledgments section or other areas of your manuscript. We will only publish funding information present in the Funding Statement section of the online submission form. 

"Funding for the WHO AFRO GIS is provided by the World Health Organization, Geneva, Switzerland, and Bill & Melinda Gates Foundation, Seattle, Washington, USA."

Response: Funding Statement should read as follows: “Funding for the WHO AFRO GIS is provided by the World Health Organization, Geneva, Switzerland, and the Bill & Melinda Gates Foundation. Editorial support was provided by Amanda Justice and Geoff Marx of BioScience Communications, New York, New York, USA, funded by the Bill & Melinda Gates Foundation.”

Response: This has already been shared and data were downloaded: https://whonghub.org/faroukh/92/1537#/table (Username: faroukh; Password: Welcome12345). Text has been updated. [Lines 371-4]

5. We note that Figures 1 to 3 in your submission contain map/satellite images which may be copyrighted. All PLOS content is published under the Creative Commons Attribution License (CC BY 4.0), which means that the manuscript, images, and Supporting Information files will be freely available online, and any third party is permitted to access, download, copy, distribute, and use these materials in any way, even commercially, with proper attribution. For these reasons, we cannot publish previously copyrighted maps or satellite images created using proprietary data, such as Google software (Google Maps, Street View, and Earth). For more information, see our copyright guidelines: http://journals.plos.org/plosone/s/licenses-and-copyright.

(1) You may seek permission from the original copyright holder of Figures 1 to 3 to publish the content specifically under the CC BY 4.0 license. 

Response: Figure 1 has been revised and does not require permission. Sources for Figures 1 and 4 have been added to legends. New Figures 2 and 5 (formerly 3) are derived from GeoPoDe, an open-source, public data repository for geospatial reference datasets, and does not require permission. However, we have requested and received permission to be safe. (Note: GeoPoDe is a public web data repository for geospatial datasets available by country. The platform allows public health officials and partner organizations to visualise geospatial data on an interactive map and download the data in multiple formats. All datasets contained in GeoPode are available to non-profit or humanitarian applications and are “operational” in nature. The datasets do not constitute authoritative, government-sanctioned reference data.)

Additional Editor Comments:

• Please review and consider all the comments and suggestions by both reviewers to provide more clarity and context within the manuscript.

Response: Done 

• Both reviewers have also highlighted the focus on Polio in the current draft and the need to revise the title and text to reflect the aim of this review as stated in Lines 92-95? Is the manuscript about GIS and Polio alone or other diseases as well? You may revise the title of the manuscript accordingly showing "past" Polio Successes." and challenges "post covid" for GIS utilization for other disease outbreaks. Any examples of how the GIS system has supported any other diseases? (how is the AFRO GIS Centre addressing these ongoing public health needs?)

Response: The title has been revised: “Geographic information system and information visualization capacity building: successful polio eradication, and current and future challenges in the COVID-19 era.”

Examples of GIS applications in routine immunization to improve vaccination coverage in the Lake Chad region,and many WHO/AFRO countries are enumerated in an earlier publication: Akpan GU et al. Conclusions of the African Regional GIS Summit (2019): using geographic information systems for public health decision-making. BMC Proc 16 (Suppl 1), 3 (2022). https://doi.org/10.1186/s12919-022-00233-y.

• The current title also includes "capacity building". The conclusion is specific for Polio and GPEI, is this capacity building for WHO GIS Polio teams or for country field staff for Polio and other diseases as well?

Response: Capacity building is for polio and other diseases, as well. As noted, title has been revised.

• In line with the recommendation on the title, please update the abstract and the conclusion to ensure alignments. Note also that the conclusion only mentions Polio and nothing about other diseases outbreaks.

Response: We have revised the last sentence in the abstract (“It also examines current and prospective challenges regarding other disease outbreaks in the COVID-19 era and how the AFRO GIS Centre is addressing these ongoing public health needs.”) to address this issue [Lines 42-3]. The end of the Conclusion has been revised: “…with the goal of faster detection and response to polio and other disease outbreaks in the COVID-19 era.” [Line 363]

• Lines 169-171 documents polio vaccination by WHO AFRO GIS teams? Does the AFRO GIS deploy vaccination teams or is this by the teams recruited by the Governments? Is there a reference for this?

Response: It is the role of WHO/AFRO to respond through vaccination, and deployment is jointly made by WHO and the respective government. Additional references have been added. https://www.afro.who.int/health-topics/polio/polio-update-on-country-situations/nigeria-and-the-lake-chad-countries;
https://polioeradication.org/countries/lake-chad-region/

• The section on use of GIS beyond Polio (Lines 245-276) is a clear example of Polio transition in action. Please include a line or two on Polio transition.

Response: We have added the following text: “The integration of polio transition processes involves the bringing together of various stakeholders and resources to ensure a smooth and effective transition. This includes integrating polio surveillance systems into existing disease surveillance networks, incorporating polio vaccinations into routine immunization programs, and merging the expertise and infrastructure developed for polio eradication with broader public health initiatives.” [Lines 291-5] 

• Please also revise to correct sentences and punctuations as may be required for consistency E.g.: Line 215 & 21: Train (Orient) participants "on" while Line 219: Train participants "in".

Response: Done

Reviewers' comments:

Reviewer's Responses to Questions

Comments to the Author

1. Is the manuscript technically sound, and do the data support the conclusions?

Reviewer #1: Partly

Reviewer #2: Partly

Response: Revisions made as specified in comments.

2. Has the statistical analysis been performed appropriately and rigorously? 

Reviewer #1: N/A

Reviewer #2: N/A

Response: N/A

3. Have the authors made all data underlying the findings in their manuscript fully available?

The PLOS Data policy requires authors to make all data underlying the findings described in their manuscript fully available without restriction, with rare exception (please refer to the Data Availability Statement in the manuscript PDF file). The data should be provided as part of the manuscript or its supporting information or deposited to a public repository. For example, in addition to summary statistics, the data points behind means, medians and variance measures should be available. If there are restrictions on publicly sharing data—e.g. participant privacy or use of data from a third party—those must be specified.

Reviewer #1: Yes

Reviewer #2: Yes

Response: Thanks 

4. Is the manuscript presented in an intelligible fashion and written in standard English?

Reviewer #1: Yes

Reviewer #2: Yes

Response: Thanks 

Reviewer #1: In summary this manuscript outlines the development and expansion of the use of GIS systems in the African WHO region for primarily polio eradication. This is a very important technological development both for polio eradication and future disease surveillance and intervention monitoring. The manuscript provides narrative of the progress to date, highlighting particular GIS gaps (i.e. it is not a scientific research article). However, to a reader not involved in this GIS network, more details are required to understand the processes. The authors also present data from a 2019 survey on national GID capacity but more details are required as to who was surveyed and whether follow-up data will be collected to measure progress. It would be much clearer if different visualizations were presented for a particular outbreak that the network has worked on e.g., The authors conclude the GIS system was important for achieving certification of WPV1 in the region, it would be helpful to see some examples of the different GIS layers that contributed to this.

Response: There are no layers involved; it was achieved through surveillance visits, with evidence of visits at the health facility level with GPS enabled on submission of visit checklists. The use of GIS enables technologies in vaccination tracking systems and the use of the AVADAR electronic surveillance checklist serves as evidence (see: https://doi.org/10.29245/2578-3009/2021/S2.1101, http://dx.doi.org/10.16966/2471-8211.151, doi: 10.11604/pamj.supp.2022.42.1.33788, 10.2196/18950), https://doi.org/10.1186/s12889-018-6187-x).

Minor comments:

Abstract:

‘Currently, 95% of reported polio cases are

---

## [Decision Letter · Decision Letter 1]

27 Jun 2024

Geographic information system and information visualization capacity building: successful polio eradication and current and future challenges in the COVID-19 era for the World Health Organization’s African region

PONE-D-23-21989R1

Dear Mr.Kipterer

We’re pleased to inform you that your manuscript has been judged scientifically suitable for publication and will be formally accepted for publication once it meets all outstanding technical requirements.

Kind regards,

Terna Ignatius Nomhwange, MD,DTM&H,MBA

Academic Editor

PLOS ONE

Additional Editor Comments (optional):

Reviewers' comments:

Reviewer's Responses to Questions

**Comments to the Author**

1. If the authors have adequately addressed your comments raised in a previous round of review and you feel that this manuscript is now acceptable for publication, you may indicate that here to bypass the “Comments to the Author” section, enter your conflict of interest statement in the “Confidential to Editor” section, and submit your "Accept" recommendation.

Reviewer #2: All comments have been addressed

2. Is the manuscript technically sound, and do the data support the conclusions?

Reviewer #2: Yes

3. Has the statistical analysis been performed appropriately and rigorously? 

Reviewer #2: N/A

4. Have the authors made all data underlying the findings in their manuscript fully available?

Reviewer #2: Yes

5. Is the manuscript presented in an intelligible fashion and written in standard English?

Reviewer #2: Yes

6. Review Comments to the Author

Reviewer #2: (No Response)

7. PLOS authors have the option to publish the peer review history of their article (what does this mean?). If published, this will include your full peer review and any attached files.

Reviewer #2: No

---

## [Editor Report · Acceptance letter]

2 Jul 2024

PONE-D-23-21989R1 

PLOS ONE

Dear Dr. Kipterer, 

I'm pleased to inform you that your manuscript has been deemed suitable for publication in PLOS ONE. Congratulations! Your manuscript is now being handed over to our production team.

Kind regards, 

on behalf of

Dr. Terna Ignatius Nomhwange 

Academic Editor

PLOS ONE